# Uncovering the Host Range for Maize Pathogen *Magnaporthiopsis maydis*

**DOI:** 10.3390/plants8080259

**Published:** 2019-07-30

**Authors:** Shlomit Dor, Ofir Degani

**Affiliations:** 1Tel-Hai College, Upper Galilee, Tel-Hai 12210, Israel; 2Migal-Galilee Research Institute, Tarshish 2, Kiryat Shmona 11016, Israel

**Keywords:** barley, cotton, fungus, host plant, late wilt, *Magnaporthiopsis maydis*, maize, quantitative real-time PCR (qPCR), *Setaria viridis*, watermelon

## Abstract

The fungus *Magnaporthiopsis maydis* is a soil-borne, seed-borne vascular wilt pathogen that causes severe damage to sensitive *Zea mays* L. (maize) hybrids throughout Egypt, Israel, India, Spain, and other countries. It can undergo virulence variations and survive as spores, sclerotia, or mycelia on plant residues. Maize, *Lupinus termis* L. (lupine) and *Gossypium hirsutum* L. (cotton) are the only known hosts of *M. maydis*. Identification of new plant hosts that can assist in the survival of the pathogen is an essential step in restricting disease outbreak and spread. Here, by field survey and growth chamber pathogenicity test, accompanied by real-time PCR analysis, the presence of the fungal DNA inside the roots of cotton (Pima cv.) plants was confirmed in infested soil. Moreover, we identified *M. maydis* in *Setaria viridis* (green foxtail) and *Citrullus lanatus* (watermelon, Malali cv.). Infected watermelon sprouts had delayed emergence and development, were shorter, and had reduced root and shoot biomass. *M. maydis* infection also affected root biomass and phenological development of cotton plants but caused only mild symptoms in green foxtail. No *M. maydis* DNA was detected in *Hordeum vulgare* (barley, Noga cv.) and the plants showed no disease symptoms except for reduced shoot weight. These findings are an important step towards uncovering the host range and endophytic behavior of *M. maydis*, encouraging expanding this evaluation to other plant species.

## 1. Introduction

Late wilt is a vascular wilt disease of *Zea mays* L. (maize) caused by the phytoparasitic fungus *Magnaporthiopsis maydis* (Samra, Sabet, and Hing; Klaubauf, Lebrun, and Crou) [1]. Synonyms are *Harpophora maydis* [2] and *Cephalosporium maydis* (Samra, Sabet, and Hingorani). Late wilt was reported in Egypt [3], India [4], Hungary [5], Romania [6], Spain and Portugal [7], Israel [8], and Nepal [9]. The pathogen can cause severe economic losses, with 80–100% infection and total yield loss reported when heavily infested fields were planted with sensitive maize hybrids [10,11].

The fungus infects the roots at an early stage of host growth, but wilt symptoms usually develop later when the plants approach the flowering stage, approximately 60 days after sowing (DAS) [10]. With disease advancement, the lower stem becomes dry (particularly at the internodes), and has a hollow and shrunken appearance, with dark yellow to brownish softened pith and brownish-black vascular bundles [12,13]. Late wilt is frequently associated with infection by secondary plant parasitic fungi causing the stem symptoms to become more severe. Indeed, several fungi—such as *Fusarium verticillioides* causing stalk rot, *Macrophomina phaseolina* causing charcoal rot, and *M. maydis*—are grouped together in a post-flowering stalk rot complex, which is considered one of the most destructive, severe, and widespread groups of diseases in maize [14].

*M. maydis* can survive and spread through infested soil, crop residues [15], seed-borne inoculum [8] or secondary hosts [16,17]. *Zea mays* (maize) and *Lupinus termis* L. (lupine, Albus cv.) are the only known hosts of *M. maydis* [17,18]. It was also reported that in *Gossypium hirsutum* L. (cotton, Bahteem 185 cv.), inoculating the soil with *M. maydis* is associated with increased production of lateral roots and the appearance of local dark red lesions and shallow cracks on young cotton roots (up to 45 DAS). Later, as cotton plants mature and their roots harden, these lesions disappear, as well as the fungus. Moreover, *M. maydis* was not recovered from these symptomatic plants [16]. *M. maydis* also causes a significant damping-off and stunting of the widely cultivated lupine in Egypt, but it is unknown if *Lupinus* spp. or other plants are secondary hosts [17].

Since the disease can appear in infested fields even after several years of growing alternative crops to maize, the identification of other plant hosts that can assist in the pathogen’s survival is an essential step in restricting inoculum build-up, disease outbreak, and spread. The current work reveals a broader host range for this phytopathogenic fungus using a field survey and growth chamber pathogenicity test accompanied by quantitative real-time PCR (qPCR) analysis.

## 2. Results

The survey of 137 plants sampled randomly from *M. maydis*-contaminated agricultural fields scattered throughout the Hula Valley in the Upper Galilee (northern Israel) or in the southern coastal plain of Israel (Yavne) was conducted during the years 2016–2017. In order to allow inoculum build-up (if present) within potential host plants, the survey was done at the end of the growing session of each field crop. In this survey, 32 cotton plants were sampled from commercial fields located in Yavne, Amir, and Menara fields. Twenty-two of those samples showed signs of dehydration, and in six of them, the qPCR analysis detected the late wilt pathogen, *M. maydis* (Table 1). The fungal pathogen could not be detected by qPCR in the other crops including 10 garlic plants from the Hulata fields; 40 pea plants from the Manara, Hulata, and Amir fields; 20 wheat plants from the Manara fields; 5 tomato plants (Niva cv.) from the Amir field; and 30 purple nutsedge (*Cyperus rotundus*) plants from the Amir and Manara fields (Table 1). The purple nutsedge is a species of weed (Cyperaceae) that is wildly distributed in the sampled commercial fields.

An experiment to identify a new host range for the late wilt pathogen was carried out under controlled conditions in a growth chamber. Five crop species were inspected as potential hosts for *M. maydis*. The selected species were all summer field crops routinely cultivated in *M. maydis* naturally infested fields in a crop rotation. After nearly five weeks of growth, *M. maydis* DNA was detected in the roots of sensitive maize (Prelude cv.), green foxtail, cotton (Pima cv.), and watermelon (Malali cv.) (Figure 1).

The levels of *M. maydis* relative DNA (*Mm*) abundance normalized to the cytochrome c oxidase (*Cox*) DNA (Figure 1A) were similar to the identification percentages (Figure 1B); green foxtail had the highest levels in both measurement methods. No DNA was detected in *Hordeum vulgare* L. (barley, Noga cv., two row verity) or the uninfected control plants. Indeed, barley had no disease symptoms except for reduced shoot weight (*p* = 0.04, Figure 2D).

Watermelon plants grown in infested soil had significantly reduced growth values compared to the control in all symptoms measurements made in this study: sprout emergence (*p* = 0.02); phenological stage (numbers of leaves, *p* = 0.05); plant height (*p* = 0.03); aboveground parts wet weight (*p* = 0.04); and root wet weight (*p* = 0.04). Infected maize plants had significantly delayed phenological development (*p* = 0.001) and reduced shoot (*p* = 0.02) and root (*p* = 0.03) biomass relative to the non-infested soil control (Figure 2B,D,E and insert). Cotton plants grown in the *M. maydis*-infested soil had a significantly decreased root biomass (*p* = 0.03, Figure 2E). In contrast, *M. maydis* infection caused only minor (insignificant) symptoms in green foxtail.

## 3. Discussion

*M. maydis* is a severe maize pathogen with limited global distribution. This study inspected several summer crop species that traditionally were grown on late wilt contaminated fields during crop rotation. These plant species were tested as potential alternate hosts of *M. maydis*. qPCR was used to identify and quantify the presence of the fungal DNA, and in addition, the fungal impact on the plant’s growth parameters was measured. Under controlled conditions in a growth chamber, the fungus, in addition to maize, could infect cotton, watermelon, and green foxtail, but not barley (Figure 1). This may hint at the possibility that barley plants have immunity (or resistance) to the pathogen. In order to prove this, other barley cultivars should be examined and a more in-depth examination should be conducted to identify the source of this immunity. In a field survey of several other plant species, the fungus was detected in cotton plants (6/32), but not in garlic, pea, wheat, tomato, and purple nutsedge (Table 1). This pioneering work should be expanded in future studies to closely examine the *M. maydis* infection impact on cotton, watermelon and green foxtail plants during a full growing season and to include additional plant species.

The results showed that *M. maydis* has the potential to infect plant species other than maize, including dicot plants. *M. maydis* is a destructive pathogen, and once introduced into a certain region of fields, it is hard to eradicate and can cause severe economic losses [19]. Therefore, the finding of this work has significant implications for risk management. Indeed, the presence of an alternative host that may assist in the pathogen’s survival during crop rotation is an important aspect to control late wilt disease. Moreover, these newly discovered hosts can also assist in spreading the pathogen to a new area.

Earlier, the interaction between *M. maydis* and *Fusarium oxysporum* on the roots of cotton and maize was investigated [16]. This interaction is of particular interest since the two fungi occur widely in Egyptian soils, and their respective hosts are cultivated alternately as summer crops in two-year rotation [16]. *L. termis* (lupine), which is widely cultivated in Egypt, is another known host for *M. maydis* [17]. Interestingly, *F. oxysporum f.* sp. lupini is also a common fungal pathogen on lupine plants in Egypt, causing wilt disease that results in severe economic losses. Since both pathogens, *M. maydis* and *F. oxysporum*, are present in diseased maize, cotton, and lupine plants [16], their interactions or cross-influence with those plants and other potential host plants may be scientifically and agriculturally important. Reviewing the literature led to the conclusion that despite its pivotal rule, *M. maydis* does not act alone; instead, it is part of a larger complex of maize pathogenic fungi (the post-flowering stalk rot complex) that includes at least two additional partners, *F. verticillioides* and *M. phaseolina* (for example, see [14]). Indeed, *F. verticillioides* is a secondary invader or opportunist that developed in late wilt-diseased, attenuated maize plants [8]. Therefore, the relationship between *M. maydis* and new host plants and their phytoparasitic pathogens has only now been revealed.

The current work is another step towards uncovering the intriguing relationship between the late wilt pathogen and a wider range of host plants. Particularly detecting *M. maydis* in green foxtail (green millet) is interesting. The fungal infection rate (frequentness, infected plants percentage) and severity (*M. maydis* DNA relative levels) in green foxtail were higher than maize, but there was no significant effect on growth or development of the green foxtail plants, and they did not develop disease symptoms (Figure 2). This species of grass is a common commercial crop in northern Israel, but it is also a wild native species in Eurasia, including Israel. However, it is known on most continents as an introduced species and is closely related to *Setaria faberi*, a noxious weed [20]. The presence of maize late wilt pathogen in this hardy grass (green foxtail)—which grows in many types of disturbed, urban, and cultivated habitats—indicates another serious risk of this pathogen spread.

The results presented here are improving the knowledge of the *M. maydis* pathogen and have epidemiological implications. The data also offer another clue to the hidden endophytic lifestyle of *M. maydis* that could be present in some hosts (such as green foxtail) without any clear visible symptoms. The current findings encourage carrying out more screening for additional potential host plants for *M. maydis* and examining the unique relationship of this fungus with each of them.

## 4. Materials and Methods

### 4.1. Field Survey

A field survey was conducted during the spring and summer of the years 2016–2017, as detailed in Table 1. The survey included 137 plants collected randomly from *M. maydis*-infested commercial fields scattered throughout the Hula Valley in the Upper Galilee (northern Israel) or in the southern coastal plain of Israel (Yavne). Different plants including *Allium sativum* (garlic), *Pisum sativum* (pea), *Triticum* sp. (wheat), *Solanum lycopersicum* (tomato, Niva cv.), and *Cyperus rotundus* (purple nutsedge) were sampled. All samples were collected at the end of the growing season right before or after the harvest. Except for one sample (22 wilted plants from the Yavne field, 20/08/2017), all plants had a healthy appearance and normal growth development. Immediately after sampling, plants were cleaned of visible soil by rinsing thoroughly under running tap water. DNAs were extracted from the root and near-surface hypocotyl tissues and analyzed using qPCR.

### 4.2. Fungal Isolates and Growth Conditions

One representative isolate of *M. maydis* called *Hm-2* (CBS 133165) was chosen for this study. This isolate is presently deposited in the CBS-KNAW Fungal Biodiversity Center, Utrecht, The Netherlands. Similar to other isolates (that can also be found in CBS-KNAW in the same collection), this *M. maydis* strain was recovered from wilting maize plants (Jubilee cv., Syngenta, Fulbourn, Cambridge, UK) sampled in Sde Nehemia (the Hula Valley, Upper Galilee, northern Israel) in 2001. These *M. maydis* isolates (including this representative strain) had previously been identified using PCR-based DNA unique fragment amplification and sequencing, and their pathogenicity, physiology, colony morphology, and microscopic traits were characterized [8,21]. The fungus was routinely grown on potato dextrose agar (PDA) (Difco, Detroit, MI, USA) at 28 ± 1 °C in dark for 4–7 days.

### 4.3. Growth Chamber Experiment

The ability of *M. maydis* to infect different hosts was tested using a sprouts assay. Sprouts were up to the age of five weeks and the assay was done in a growth chamber. The sweet maize Prelude cv. (from SRS Snowy River seeds, Australia, provided by Green 2000 Ltd., Bitan Aharon, Israel) was used in this experiment because of its susceptibility to late wilt [10]. Other summer field crops tested in this work as being potential *M. maydis* hosts were cotton (Pima cv.), *Setaria viridis* (common name green foxtail or green millet), *Citrullus lanatus* (watermelon, Malali cv.), and *Hordeum vulgare* (barley, Noga cv.). We used an inoculum method consisting of naturally infested soil from infested maize field. The soil was collected randomly from several places in a field (Mehogi 5 maize plot) near Kibbutz Amir (the Hula Valley, Upper Galilee, northern Israel) from the top 15 cm of the soil surface (the pathogen is restricted to the top 20 cm of soil [15]). The naturally infested soil was mixed with 30% Perlite no. 4. Even in heavily infested fields, the spreading of the pathogen is not uniform. The pathogen is scattered in the soil and the spreading of the disease is not uniform in the field. Thus, we used complementary inoculum to ensure inoculation of the plants as high and uniform as possible. The complementary inoculation with the *Hm-2* isolate was done in two steps as follows [22]. First, 20 g of autoclave-sterilized wheat seeds were incubated for 20 days or more with *M. maydis* culture agar disks. The *M. maydis* disks (6-mm-diameter) were cut from the margins of young (3–5-day-old) *M. maydis* colonies grown on PDA at 28 ± 1 °C in darkness. Inoculated seeds were mixed with the top 20 cm of soil 2–3 days before sowing. These wheat seeds were only used to spread the pathogen in the soil. Control treatment was sterilized wheat seeds incubated in the same conditions without the fungus. Second, fungal disks (prepared as mentioned above) were added to each seedling when the plants first emerge above the ground surface. To carry out this procedure with minimal interference in the sprout’s development, a glass tube (10 × 75 mm) was placed next to each seed during sowing. On the inoculation day (7–10 days from seeding when the plants first appeared above the ground surface), the glass tubes were taken out and two fungal disks were added (approximately 4 cm below the ground surface). The remaining cavity was filled with soil. The non-inoculated control plants were grown under the same conditions.

Each treatment included five independent replications (pots). The experiment was conducted in triplicate (with similar results). Thus, 15 plants of each species were tested in total. Each experiment was carried out in a completely randomized design. Five maize seeds were sown in a 2.5-L pot about 4 cm beneath the surface and grown in a growth chamber under a constant temperature of 28 ± 3 °C, a relative humidity of 45–50% and a 12-h photoperiod. Watering was done by adding 100 ± 10 mL water every 72 h to the pots using a computerized irrigation system. Emergence degree was evaluated at 10 DAS. The plant’s phenological stage, fresh roots and aboveground biomass and height assessment were measured 37 DAS. At the experiment end, tissues were sampled from the root and near-surface hypocotyl, and processed for DNA purification and analyzed using qPCR, as detailed below

### 4.4. Molecular Diagnosis of Magnaporthiopsis maydis

The molecular diagnosis was performed on plants collected from the field and those from the growth chamber experiment. Roots were washed under tap water to remove soil. Tissues were sampled by cross-sectioning about 2 cm in length from each root; total weight of each tissue was 0.7 g and considered one replication. Tissue samples were moved to universal extraction bags (Bioreba, Reinach, Switzerland) with 4 mL CTAB buffer and were ground with a manual tissue homogenizer (Bioreba, Reinach, Switzerland) for 5 min until the tissues were completely homogenized. The homogenized samples were treated for DNA purification, as described below.

Total DNA was isolated from the maize tissue samples using the procedure of [23] with slight modifications. After grinding the tissue samples with 4 mL buffer (0.7 M NaCl, 1% cetyltriammonium bromide (CTAB) buffer, 50 mM Tris-HC1 pH 8.8, 10 mM EDTA and 1% 2-mercaptoethanol), 1.2 mL from the blend was maintained for 20 min at 65 °C. For DNA purification, we used the Eppendorf Centrifuge 5810 R. The samples were centrifuged at room temperature (25 °C) at 14,000 rpm for 5 min. The upper part of the lysate (usually 700 µL) was then extracted with an equal volume of chloroform/isoamyl alcohol (24:1) After mixing by vortex, the mixture was centrifuged again at 14,000 rpm for 5 min at room temperature (25 °C). The chloroform/isoamyl-alcohol extraction was repeated twice. The supernatant (usually 300 µL) was then transferred to a new Eppendorf tube and mixed with cold isopropanol (2:3). The DNA solution was gently mixed by inverting the tube several times, kept for 20–60 min at minus 20 °C, and centrifuged (14,000 rpm for 20 min in 4 °C). The precipitated DNA isolated was resuspended in 0.5 mL 70% ethanol. After additional centrifugation (14,000 rpm at 4 °C for 10 min), the DNA pellet was dried in a sterile hood overnight. Eventually, the DNA was suspended in 100 µL HPLC-grade water and kept at minus 20 °C until use.

All of the qPCR reactions were performed as previously described [10] using the ABI PRISM^®^ 7900 HT Sequence Detection System (Applied Biosystems, Foster City, CA, USA) for 384-well plates. The 5 µL of total reaction was used per sample well including 2 µL of sample DNA extract, 2.5 µL iTaq™ Universal SYBR^®^ Green Supermix (Bio-Rad Laboratories Ltd., Rishon Le Zion, Israel), 0.25 µL forward primer, and 0.25 µL reverse primer (10 µM from each primer to a well). The qPCR program was as follows: precycle activation stage, 1 min at 95 °C; 40 cycles of denaturation (15 s at 95 °C), and annealing and extension (30 s at 60 °C), followed by melting curve analysis. Plant samples (root and stem tissues) from each experiment were analyzed separately by qPCR. Each sample was tested four times by qPCR to ensure consistency of the results. The A200a primers were utilized for qPCR (sequences in Table 2). The gene encoding for the last enzyme in the respiratory electron transport chain of the eukaryotic mitochondria-cytochrome c oxidase (*COX*)-was used as a “housekeeping” reference gene to normalize the amount of DNA [24]. This gene was amplified using the COX F/R primer set (Table 2). Proportionate quantification of the target *M. maydis* fungal DNA was calculated according to the ΔΔCt model [25]. Efficiency was supposed to be the same for all samples. All amplifications were performed in triplicate.

### 4.5. Statistical Analyses

In all experiments we used fully randomized statistical designs. To evaluate the *M. maydis* infection outcome on symptoms in the growth chamber experiment, the Student’s *t*-test (with a significance threshold of *p* = 0.05) was used. This test compared the treatment means to the control.

## Figures and Tables

**Figure 1 plants-08-00259-f001:**
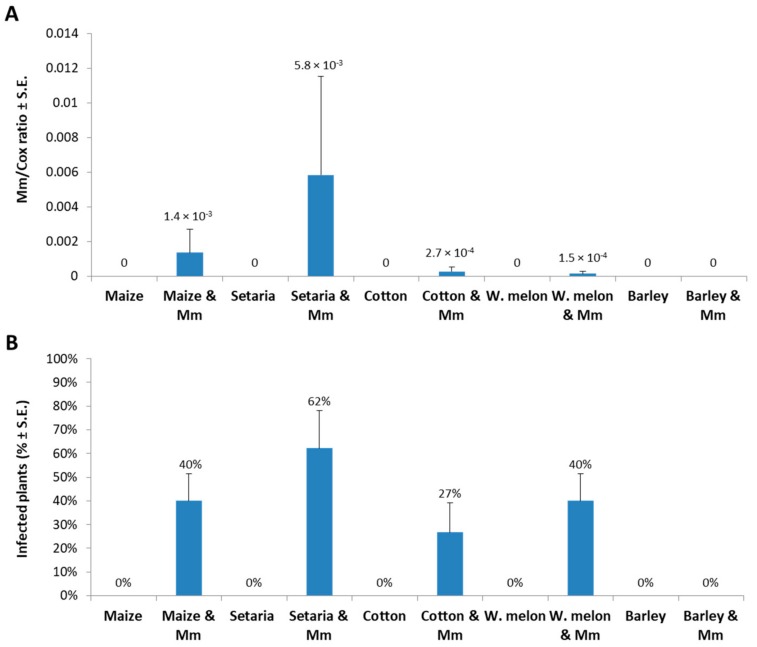
qPCR diagnosis of *Magnaporthiopsis maydis* infection in growth chamber. Five crop species were analyzed using qPCR to identify the ability of *M. maydis* to infect and colonize the root tissues 37 days after sowing. These species included maize, *Setaria viridis* (common name green foxtail or green millet), cotton, watermelon (W.melon), and barley. (**A**) *M. maydis* relative DNA (*Mm*) abundance normalized to the cytochrome c oxidase (*Cox*) DNA. (**B**) The percentage of infected plants (with *M. maydis* DNA inside their tissues) identified by the qPCR molecular method. Controls-uninfected plants examined in those experiments (all had zero levels of *M. maydis* relative DNA). Bars indicate an average of three independent experiments; each included five replications (pots). Standard errors are indicated in error bars.

**Figure 2 plants-08-00259-f002:**
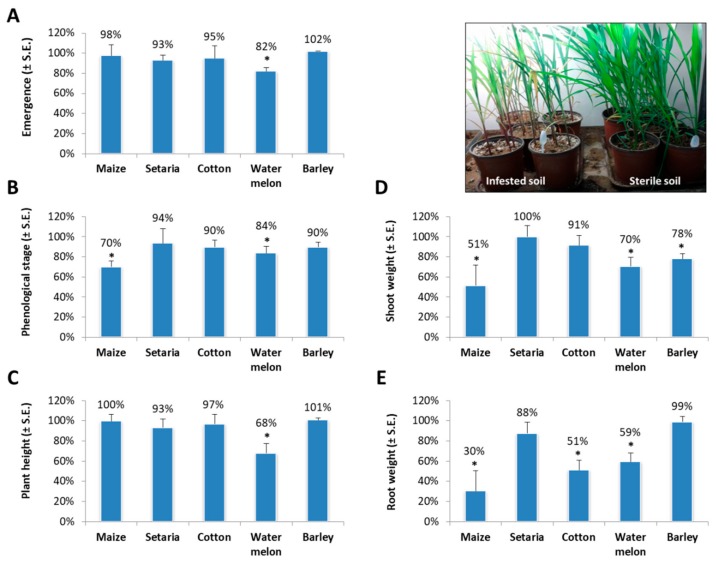
Sprout pathogenicity assay. Symptoms evaluation was made for the cultivar species inspected in Figure 1. Values are expressed as being different from the control (uninfected plants) in percentages. Sprout emergence (**A**), phenological stage (numbers of leaves, (**B**), plant height (**C**), aboveground parts wet weight (**D**), and root wet weight (**E**) were determined after growing the different plant species in a soil infected with *M. maydis* in a growth chamber for 37 days. The results are the average of two separate experiments; each included five independent replications (pots). Vertical upper bars represent the standard error. Significance from the control (untreated plants) is indicated as * = *p* < 0.05. Insert—photo of the sweet, sensitive maize Prelude cv. at the end of the experiment.

**Table 1 plants-08-00259-t001:** Random sampling from *Magnaporthiopsis maydis*-contaminated agricultural fields scattered throughout the Hula Valley in the Upper Galilee (northern Israel) or in Yavne, in the southern coastal plain of Israel ^1^.

Crop	Location	Date	Growing Stage and Health	Sample Size (Plants)	qPCR *M. maydis* Detection
Cotton	Yavne	28/07/2016	End session-healthy	5	-
		28/07/2016	End session-diseased	5	-
		20/08/2017	End session-diseased	22	6
Garlic	Hulata	05/04/2016	End session-healthy	10	-
Pea	Manara	05/04/2016	End session-crop residues	10	-
	Hulata	05/04/2016	End session-healthy	10	-
	Amir	21/06/2016	End session-healthy	20	-
Wheat	Manara	05/04/2016	End session-post harvest	20	-
Tomato (Niva)	Amir	18/05/2016	End session-healthy	5	-
*Cyperus rotundus* (purple nutsedge)	Manara	05/04/2016	End session-post harvest	10	-
	Amir	02/08/2016	End session-healthy	10	-
	Amir	21/06/2017	End session-healthy	10	-

^1^ A survey of 137 plants randomly sampled from *M. maydis*-contaminated agricultural fields. All the plants were collected at the end of the growth session right before or after the harvest. Except for one sample (22 wilted plants from the Yavne field, 20/08/2017), all plants had a healthy appearance and normal growth development.

**Table 2 plants-08-00259-t002:** Primers for *Magnaporthiopsis maydis* qPCR detection ^1.^

Pairs	Primer	Sequence	Uses	Amplification	References
**Pair 1**	A200a-for A200a-rev	5′-CCGACGCCTAAAATACAGGA-3′ 5′-GGGCTTTTTAGGGCCTTTTT-3′	Target gene	*M maydis* AFLP-derived species-specific fragment	[8]
**Pair 2**	COX-F COX-R	5′-GTATGCCACGTCGCATTCCAGA-3′ 5′-CAACTACGGATATATAAGRRCCRRAACTG-3′	Control	Cytochrome c oxidase (COX) gene product	[24,26]

^1^ qPCR - quantitative real-time PCR; and AFLP—amplified fragment length polymorphism.

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
