# Peer review of "Uncovering the Host Range for Maize Pathogen Magnaporthiopsis maydis"

_plants, 2019, doi:10.3390/plants8080259_

Round 1

Reviewer 1 Report

This paper is about finding new potential hosts of Magnaporthiopsis maydis. Followings are some points to improve the manuscript:

Comments:

1- What is the statistical analysis used in this work? Please add it to methods.

2- Both tables in manuscript are labeled as Table 1! It should be corrected. 

3- When use a name for first time in manuscript it should be written in full, but after that use the acronym. Please revise the manuscript to make sure that consistency is maintained. 

4- In methods, section 4.3 is not very clear! If infested soil was used, why inoculated wheat seeds are mixed again with soil? And also authors did not talk about other plants they used in this experiment. 

5- Most parts of manuscript, specifically in methods, is overlapped with authors’ previous publications. This is not acceptable, so it is better to rephrase those parts. 

6- Authors can find more comments in attached file.

Author Response

Responses to Reviewer 1 comments

We thank the reviewer for investing substantial work, which contributed greatly to this manuscript. The many helpful and important remarks and suggestions improved this scientific paper remarkably and made it more accurate, clear, focused and well-structured.

We have highlighted the revisions in the updated version to address your and the other reviewers’ suggestions and remarks. We are also providing a clean copy of the revised manuscript that includes all the corrections embedded in the text.

What is the statistical analysis used in this work? Please add it to methods.

We chose to do a series of Student’s T tests (which is essentially the same as the unprotected Fisher's LSD test – a set of T tests, without any correction made for multiple comparisons), because our greatest interest was to discover differences between two groups – each treatment compared separately to the control. To this end, the Student’s T test can provide a more powerful test (i.e., enabling more easily to accept the test alternative hypothesis – finding differences). The following section was added to the text (Lines 275-278):  

“4.5 Statistical analyses

In all experiments we used fully randomized statistical designs. To evaluate the M. maydis infection outcome on symptoms in the growth chamber experiment, the Student’s t-test (with a significance threshold of P = 0.05) was used. This test compared the treatment means to the control.”

Both tables in the manuscript are labeled as Table 1! It should be corrected.

Table 2 was labeled correctly as advised.

When using a name for first time in the manuscript it should be written in full, but after that use the acronym. Please revise the manuscript to make sure that consistency is maintained.

The reviewer is correct. All organisms’ names in the manuscript were revised to ensure that the name is written in full, the first time it appears in the manuscript, and as an acronym afterwards.

In methods, section 4.3 is not very clear! If infested soil was used, why inoculated wheat seeds are mixed again with soil? And also the authors did not talk about other plants they used in this experiment.

Naturally infested soil taken from a maize field was chosen for the growth chamber pathogenicity trials. However, we cannot rely on natural soil infestation alone, which can lead to highly variable results. Even in heavily infested fields, the spreading of the pathogen is not uniform. The pathogen is scattered in small quantities in the soil and the disease spreading is not uniform in the field (see, for example, Degani et al., Methods for studying Magnaporthiopsis maydis, the maize late wilt causal agent. Agronomy 2019, 9, 181, Figure 1).

Thus, we used a complementary inoculum to ensure inoculation of the plants as high and uniform as possible. The difference between this method and the use of artificial soil inoculation (adding inoculum to non-contaminated soil), was discussed in our previous work (see Degani et al., Evaluating Azoxystrobin seed coating against maize late wilt disease using a sensitive qPCR-based method. Plant Disease (2019), 103 (2) 238-248). Deliberately infecting healthy soil with M. maydis was incapable of causing significant symptoms in mature plants (aged 70 days or more).

The following explanation was added to the text (lines 206-208): “Even in heavily infested fields, the spreading of the pathogen is not uniform. The pathogen is scattered in the soil and the spreading of the disease is not uniform in the field. Thus, we used complementary inoculum to ensure inoculation of the plants as high and uniform as possible.”

All plant species used in this experiment are detailed and were grown in the same inoculated soil (naturally infested soil with complementary inoculation). As written in the manuscript (Lines 213-214), the autoclave-sterilized wheat seeds were only used to spread the pathogen in the soil.

Most parts of the manuscript, specifically in methods, are overlapped with authors’ previous publications. This is not acceptable, so it is better to rephrase those parts.

We made our best effort to address this issue. Indeed, we had used the successful methods reported earlier by our group (Degani et al., Methods for studying Magnaporthiopsis maydis, the maize late wilt causal agent. Agronomy 2019, 9, 181) and referred to them in the text. Together with this, the reviewer is correct and the text should be rephrased carefully to ensure no overlap. We apologize for this mistake. English is not our native language and we tend to be somewhat conservative in writing and using repetitive vocabulary. We used the Grammarly program (San Francisco, CA, USA) to ensure that no similarities exist between this and other works. We revised the entire manuscript and replaced all of the paragraphs that had included similarities in writing style.

Response to comments in the attached file.

We accepted all the reviewer’s suggestions to delete unnecessary words and replaced them with other words to improve the sentences.

Line 102: What is the statistical analysis used in this work? Please mention it in methods.

As explained in our response in section 1, we used the Student’s T test for assessing the M. maydis infection outcome on symptoms in the growth chamber experiment. Section 4.5 with statistical analyses was added to the Materials and Methods section.

Line 110: What is INSERT?

The word “INSERT” is commonly used to refer to a graphical object (for example, a photo) embedded within another graphical object (for example, a graph). In Figure 2, we used a photograph of the maize plants experiment as an INSERT to exemplify the late wilt disease outcome on the maize sprouts development.

Line 160: Do not repeat full name.

Corrected as advised.

Line 173: There are two Table 1 in the manuscript!

Corrected as advised.

Line 206: It is not clear. Please rewrite the sentence.

The entire paragraph (lines 207-220) was edited, rewritten and expanded to clarify the soil inoculation method.

Line 207: This is not clear. If the authors used infested soil, why did they inoculate wheat seeds and mix it with soil again? Mixed soil with inoculated seeds used for what?

In our response in section 4, we explained in detail the reason for using complementary inoculum. We also added a thorough explanation to the text (lines 206-208).

Line 264: Authors should use full name of qPCR when mentioning it the first time in the manuscript.

The reviewer is right. We corrected this (lines 57-59): “The current work reveals a broader host range for this phytopathogenic fungus using a field survey and growth chamber pathogenicity test accompanied by quantitative real-time PCR (qPCR) analysis.”

Reviewer 2 Report

The study is interesting and is a way of applying what was previously published in Agronomy 2019.The results shown suggest that there is great uncertainty about the potential of this phytopathogenic infection to other plants.However, the identification of the genes from M maydis by molecular biology methods are strong but expensive since the infection is difficult to identify although very harmful.As a short communication, it is interesting to report on the usefulness of molecular methods to identify a phytopathogen that is so difficult to identify. Adjusting the discussion towards the selection criteria of the plants would be a great contribution.

Author Response

Responses to Reviewer 2 comments

We would like to express our appreciation to the reviewer for the important and helpful corrections, suggestions and advice. This contribution significantly improved the manuscript. Thank you.

We have highlighted the revisions in the updated version to address your and the other reviewers’ suggestions and remarks. We are also providing a clean copy of the revised manuscript that includes all the corrections embedded in the text.

Line 124: What was the criterion to choose those other species?

This was explained in the results section (lines 81-82): “The selected species were all summer field crops routinely cultivated in M. maydis naturally infested fields in a crop rotation.” We added the following explanation to the discussion as well (lines 121-123): “This study inspected several summer crop species that traditionally were grown on late wilt contaminated fields during crop rotation. These plant species were tested as potential alternate hosts of M. maydis.”

Line 127: this plant is resistant?

Indeed, apparently the barley plants were not infected with the pathogen and the M. maydis contaminated soil had no effect on its growth parameters (except for reduced shoot weight). This was already stated in the Results section (lines 100-101). This may hint at the possibility that barley plants have immunity (or resistance) to the pathogen. In order to prove this, other barley cultivars should be examined and a more in-depth examination should be conducted to identify the source of this immunity. This explanation was added to the discussion (lines 127-129).

Line 145: These two pathogens can survive together during the infection or is one dominant?

That is good a question. As far as we know, they can indeed exist together during the infection, but M. maydis is the dominant pathogen in maize late wilt disease and Fusarium spp. is a secondary parasite that develops in the weakening diseased plants. We explained this in our recent report (Degani et al., Methods for studying Magnaporthiopsis maydis, the maize late wilt causal agent. Agronomy 2019, 9, 181): “During the 1990s in Israel, it was assumed that the phenomenon of wilting of maize plants in commercial fields was the result of Fusarium verticillioides infestation since this pathogen was the most abundant in the infested plant samples. It was eventually proven by Koch’s postulates that the direct cause of the wilting is M. maydis, while F. verticillioides is a secondary invader or opportunist that developed in these attenuated maize plants (Drori et al., O. Molecular diagnosis for Harpophora Maydis, the cause of maize late wilt in Israel. Phytopathol. Mediterr. 2013, 52, 16-29).” An explanation was added to the text (lines 152-153): “Indeed, F. verticillioides is a secondary invader or opportunist that developed in late wilt-diseased, attenuated maize plants [8].”

Line 152: This point is important since it suggests interactions between the pathogens and the type of plant.

Indeed, this is the main statement of this research.

Line 154: The plant-fungus interaction is interesting, since from this study it is translated that the signals sent by the plants can protect or not from the phytopathogen.

We agree, this is an excellent point. However, this assumption should be verified and established in a follow-up work.

Reviewer 3 Report

Review of short communication manuscript for Plants titled “Uncovering the host range for maize pathogen Magnaporthiopsis maydis

The manuscript is generally well-written, and the authors conducted research that is important for identifying hosts of the fungus that can be useful for its control. The following minor corrections are suggested to improve the manuscript

L14 Indicate as Zea mays L. (maize), delete “apparently”

L30” Use either corn or maize

L46: Write as Zea mays since the botanical name is used for lupine

L66: Move the following to the end of the sentence to improve clarity: “We also tested for the presence of the pathogen’s DNA, with no positive results”

L74-76: Delete “throughout the Hula Valley in the Upper Galilee (northern Israel), or in the Israel southern coastal plain 75 (Yavne), was conducted during the years 2016-2017” and leave for the methods section

Figure 1: Use "&" instead of "+" and place below (i.e. on the same line as Mm. Right now, it appears as Maize Mm + Sitaria;

Indicate as percentage of infected plants;

Delete “Five 86 cultivar species were evaluated under controlled conditions to identify the ability of M. maydis to infect and colonize potential host tissues. These species included maize (sweet, sensitive maize Prelude cv.), Setaria viridis (common name green foxtail or green millet), cotton (Pima cv.), watermelon (Malali cv.) and barley (Noga cv.). Plants were harvested on 37 DAS. DNA was extracted from the plant roots and analyzed using qPCR, performed to amplify a specific M. maydis segment.

L103: Indicate as Figure 2D

L110: Move “Insert – photo of the sweet, sensitive maize Prelude cv. at the end of 110 the experiment.: to the end of L113

L112: Delete “when existing”

L145: Insert a comma between maize and cotton

L156: indicate as fungus’ infection

L212: Delete ‘complete’

L244: Indicate details of the centrifuge used or convert to x g rather than rpm and correct same in other places

Author Response

Responses to Reviewer 3 comments

We would like to express our appreciation to the reviewer for the important and helpful corrections, suggestions and advice. This contribution significantly improved the manuscript. Thank you.

We have highlighted the revisions in the updated version to address your and the other reviewers’ suggestions and remarks. We are also providing a clean copy of the revised manuscript that includes all the corrections embedded in the text.

L14 Indicate as Zea mays L. (maize), delete “apparently”

Deleted as advised.

L30” Use either corn or maize

Corrected to “maize” as advised. We used the term “maize” throughout the text.

L46: Write as Zea mays since the botanical name is used for lupine

Corrected as advised.

L66: Move the following to the end of the sentence to improve clarity: “We also tested for the presence of the pathogen’s DNA, with no positive results”

The sentence was modified and rewritten according to the suggestion by Reviewer 1. It now reads: “The fungal pathogen could not be detected by qPCR in the other crops including 10 garlic plants from the Hulata fields, 40 pea plants from the Manara, Hulata and Amir fields, 20 wheat plants from the Manara fields, five tomato plants (Niva cv.) from the Amir field, and 30 purple nutsedge plants from the Amir and Manara fields (Table 1).” (lines 68-71).

L74-76: Delete “throughout the Hula Valley in the Upper Galilee (northern Israel), or in the Israel southern coastal plain 75 (Yavne), was conducted during the years 2016-2017” and leave for the methods section

Deleted as advised.

Figure 1: Use “&” instead of “+” and place below (i.e. on the same line as Mm. Right now, it appears as Maize Mm + Sitaria; Indicate as percentage of infected plants

In Figure 1, we replaced “+” with “&” as suggested by the reviewer. However, it was not possible to put each of the bar graphs labels on the same line while maintaining the same font size. This could be modified later according to the journal style during the manuscript production stage.

Delete “Five cultivar species were evaluated under controlled conditions to identify the ability of M. maydis to infect and colonize potential host tissues. These species included maize (sweet, sensitive maize Prelude cv.), Setaria viridis (common name green foxtail or green millet), cotton (Pima cv.), watermelon (Malali cv.) and barley (Noga cv.). Plants were harvested on 37 DAS. DNA was extracted from the plant roots and analyzed using qPCR, performed to amplify a specific M. maydis segment.

We edited and significantly shortened this paragraph in the Figure 1 legend. However, we advise keeping minimal information that we think is necessary. The paragraph is now written (lines 87-90): “Five crop species were analyzed using qPCR to identify the ability of M. maydis to infect and colonize the root tissues 37 days after sowing. These species included maize, Setaria viridis (common name green foxtail or green millet), cotton, watermelon and barley.”

L103: Indicate as Figure 2D

Corrected as advised.

L110: Move “Insert – photo of the sweet, sensitive maize Prelude cv. at the end of 110 the experiment.: to the end of L113

Corrected as advised.

L112: Delete “when existing”

Deleted as advised.

L145: Insert a comma between maize and cotton

Corrected as advised.

L156: Indicate as fungus’ infection

Corrected as advised.

L212: Delete ‘complete’

Deleted as advised.

L244: Indicate details of the centrifuge used or convert to x g rather than rpm and correct same in other places

For DNA purification, we used the Eppendorf Centrifuge 5810 R. This information was added to the text (line 245).

Round 2

Reviewer 2 Report

Authors answered correctly to my questions. This study improves  knowledge to the behavior of pathogenic fungi.